

# Identification of dysregulated genes in rheumatoid arthritis based on bioinformatics analysis

Ruihu Hao[1,*], Haiwei Du[2,*], Lin Guo[1], Fengde Tian[1], Ning An[1], Tiejun Yang[3], Changcheng Wang[1], Bo Wang[1] and Zihao Zhou[1]

[1] Department of Orthopedics, Affiliated Zhongshan Hospital of Dalian University, Dalian, China
[2] Department of Bioinformatics, Beijing Medintell Biomed Co., Ltd, Beijing, China
[3] Department of Orthopedics, Affiliated Hospital of BeiHua University, Jilin, China
[*] These authors contributed equally to this work.

Corresponding author
Lin Guo, lin_guo736@163.com

## ABSTRACT

**Background**. Rheumatoid arthritis (RA) is a chronic auto-inflammatory disorder of joints. The present study aimed to identify the key genes in RA for better understanding the underlying mechanisms of RA.

**Methods**. The integrated analysis of expression profiling was conducted to identify differentially expressed genes (DEGs) in RA. Moreover, functional annotation, protein–protein interaction (PPI) network and transcription factor (TF) regulatory network construction were applied for exploring the potential biological roles of DEGs in RA. In addition, the expression level of identified candidate DEGs was preliminarily detected in peripheral blood cells of RA patients in the GSE17755 dataset. Quantitative real-time polymerase chain reaction (qRT-PCR) was conducted to validate the expression levels of identified DEGs in RA.

**Results**. A total of 378 DEGs, including 202 up- and 176 down-regulated genes, were identified in synovial tissues of RA patients compared with healthy controls. DEGs were significantly enriched in axon guidance, RNA transport and MAPK signaling pathway. RBFOX2, LCK and SERBP1 were the hub proteins in the PPI network. In the TF-target gene network, RBFOX2, POU6F1, WIPF1 and PFKFB3 had the high connectivity with TFs. The expression status of 11 candidate DEGs was detected in GSE17755, the expression levels of MAT2A and NSA2 were significantly down-regulated and CD47 had the up-regulated tendency in peripheral blood cells of patients with RA compared with healthy individuals. qRT-PCR results of MAT2A, NSA2, CD47 were compatible with our bioinformatics analyses.

**Discussion**. Our study might provide valuable information for exploring the pathogenesis mechanism of RA and identifying the potential biomarkers for RA diagnosis.

## INTRODUCTION

Rheumatoid arthritis (RA) is a chronic autoinflammatory disorder of joints, which is characterized by irreversible joint damage and destruction of cartilage and bone, leading

to severe disability or even premature death (*Aletaha et al., 2010*). It affects approximately 1% of the population worldwide, and reduces life expectancy of patients by 3–18 years. There are 25–50 new cases in a population of 100,000 worldwide each year (*Gibofsky, 2014*; *Pincus & Callahan, 1986*; *Michelsen et al., 2015*).

The etiology and pathophysiology of RA are not well understood (*McInnes & Schett, 2011*). It is widely acknowledged that RA is triggered by the complex interaction among genetic susceptibility, autoimmune disorders and dysregulated genes.

For genetic susceptibility, several studies have demonstrated that expression quantitative trait loci (eQTL) single nucleotide polymorphisms (SNPs) increase the risk of RA. A recent study indicates that *HLA-DOA* rs369150-A polymorphism reduces the expression of *HLA-DOA*, which independently associates with RA risk in the Japanese population, Asian population (Chinese and Korean) and European population (*Okada et al., 2016*). eQTL of GZMB identifies higher expression in the presence of the minor allele of rs8192916, which is associated with progression of joint destruction in RA (*Knevel et al., 2013*). For dysregulated genes, down-regulation of miR-126 inhibits PI3K/AKT signaling pathway and disrupts the imbalance between cell growth and cell death of rheumatoid arthritis synovial fibro-blasts by targeting PIK3R2 (*Gao et al., 2016*). Over-expression of TLR-2 and IL-6 decrease the expression of FCRL4, might contribute to the RA process (*Khanzadeh et al., 2016*; *Eser & Sahin, 2016*).

Previous studies demonstrate that early diagnosis and treatment will be beneficial to reduction of progressive joint injury of patients with RA. Currently, several biomarkers such as citrullinated proteins, rheumatoid factor, S100 proteins, matrix metalloproteinases and serum amyloid proteins have been described (*Smolen, Aletaha & McInnes, 2016*), but these biomarkers have limitations on sensitivity and or specificity in routine clinical practice. The early diagnosis of RA remains challenging.

In this study, we used bioinformatics methods to integrate mRNA expression datasets, which were available in the Gene Expression Omnibus database, identified DEGs in synovial tissues of patients with RA and healthy individuals, aiming to investigate the potential pathogenesis mechanism of RA and provide valuable information for the identification of potential diagnosis biomarkers in RA.

## MATERIALS AND METHODS

### Data collection

NCBI Gene Expression Omnibus (GEO) database provides a large collection of microarray expression data (*Edgar, Domrachev & Lash, 2002*). "Rheumatoid Arthritis" was used as the key word to search expression profiling of RA. The inclusion criteria of datasets were set as follows: (1) whole genome expression data of RA; (2) the expression profiling of synovial tissues was generated from patients with RA and healthy individuals.

Total of nine mRNA expression datasets of synovial tissues of RA were incorporated into our study. Four datasets were based on GPL570 [HG-U133_Plus_2] Affymetrix Human Genome U133 Plus 2.0 Array, four datasets were based on GPL96 [HG-U133A] Affymetrix Human Genome U133A Array, and one dataset was based on GPL91 [HG_U95A] Affymetrix Human Genome U95A Array.

## Data preprocessing and Identification of DEGs

The raw expression datasets were downloaded and preprocessed by log2 transformation in R language (*Shi et al., 2006*). The Linear Models "limma" package in R language was used to analyze the microarray datasets (*Smyth, 2005*). Differentially expressed genes were identified in patients with RA compared to healthy individuals. The false discovery rate (FDR) (*Reiner-Benaim, 2007*) was utilized for multiple testing corrections by using the Benjamini and Hochberg method. FDR < 0.05 was set as the threshold of DEGs.

## Heatmap analysis

To assess the similarity of gene expression patterns between two samples, two-way hierarchical clustering analysis (*Cao, 2016*) was performed. The heatmap was constructed using 'pheatmap' package in R language.

## Functional annotation of DEGs

The significantly dysregulated genes between patients with RA and healthy individuals were filtered as DEGs. In order to understand the biological roles of DEGs, the Kyoto Encyclopedia of Genes and Genomes (KEGG) pathway and Gene Ontology (GO) terms were enriched by the online tool GeneCoDis3 (*Kanehisa, 2002*; *Carmona-Saez et al., 2007*; *Torto-Alalibo et al., 2014*). The threshold of GO function and KEGG pathway enrichment were FDR < 0.05.

## Protein–protein interaction network

In order to explore the interaction between the top 10 up- and down-regulated DEGs in synovial tissues of patients with RA, BioGRID, a database of known protein interactions, was used to predict the PPI association among DEGs and the PPI interaction network was visualized by Cytoscape (*Shannon et al., 2003*; *Chatr-Aryamontri et al., 2015*).

## Construction of transcription factors regulatory network

TRANSFAC (*Wingender et al., 1996*) is a database covering details of TFs and their DNA binding sites. In our analysis, the TFs in the human genome and the motifs of genomic binding sites were downloaded from the TRANSFAC. Moreover, the 2 KB sequence in the upstream promoter region of top 10 up- and down-regulated DEGs was downloaded from UCSC (http://www.genome.ucsc.edu/cgi-bin/hgTables). Target sites of potential TFs were then distinguished. The TFs and its target genes were used for constructing regulatory network, and the network was visualized by Cytoscape software (http://cytoscape.org/) (*Shannon et al., 2003*). Nodes represented TFs or target genes and lines represented association between TFs and target genes.

## The DEGs were analyzed in the GSE17755 datasets

We obtained a number of key DEGs in RA synovial tissues based on the bioinformatics analyses. In order to analyze whether those DEGs were dysregulated in peripheral blood cells of patients with RA, the expression levels of representative DEGs were preliminarily validated in the expression profiling of peripheral blood cells of patients with RA from the publicly available GEO dataset. "Rheumatoid arthritis" was used as the key word to search the expression profiling in the GEO database. The mRNA expression profiling datasets were
included based on the following criteria: (1) the dataset was generated from the peripheral blood cells of patients with RA; (2) The expression profiling of peripheral blood cells of RA patients and healthy individuals were available in the dataset; (3) the RA patients had no treatment record before peripheral blood cells collection. Finally, GSE17755 and GSE15573 were left. A total of 18 patients with RA and 15 healthy individuals were included in the GSE15573 datasets, which had the smaller sample size compared to GSE17755. There were 112 patients with RA and 45 healthy individuals in GSE17755 dataset. Based on aforementioned information, GSE17755 was incorporated into ours study (*Lee et al., 2011*). Box-plot analysis was performed to preliminarily detect the expression levels of DEGs in peripheral blood cells of RA patients and health individuals.

### Quantitative real-time polymerase chain reaction (qRT-PCR)

In order to validate the expression levels of candidate genes in RA, 10 peripheral blood samples were collected from RA patients, who were diagnosed at the Affiliated Zhongshan Hospital. Moreover, 10 matched peripheral blood samples were collected from healthy individuals. This work was approved by the Ethics Committee of the Affiliated Zhonghan Hospital and informed written consent was obtained from all patients. The research complied with the principles of the Declaration of Helsinki.

Total RNA of peripheral blood samples were extracted by using Trizol (Invitrogen, Carlbad, CA, USA) according to the manufacture instructions. FastQuant cDNA Synthesis Kit (Tiangen, Beijing, China) was used to synthesize the cDNA of mRNA. qRT-PCR reactions were performed by using SuperReal PreMix Plus SYBR Green Kit (Tiangen, Beijing, China) on Applied Biosystems 7500 (Applied Biosystems, Foster City, CA, USA). GAPDH were used as internal control for mRNA detection. The PCR primers used in our study were shown as follows: GAPDH forward primer 5′ GGA GCG AGA TCC CTC CAA AAT 3′, reverse primer 5′ GGC TGT TGT CAT ACT TCT CAT GG 3′; MAT2A forward primer 5′ ATG AAC GGA CAG CTC AAC GG 3′, reverse primer 5′ CCA GCA AGA AGG ATC ATT CCA G 3′; NSA2 forward primer 5′ CAC CGT AAA CGC TAT GGA TAC C 3′, reverse primer 5′ GCT AGG CCT TCA GAC CAA TCA TT 3′; CD47 forward primer 5′ TCC GGT GGT ATG GAT GAG AAA 3′, reverse primer 5′ ACC AAG GCC AGT AGC ATT CTT 3′. At least triple experiments were subjected to qRT-PCR verification. The relative expression of candidate genes was calculated by using the $\Delta CT$ equation methods. Mean $\pm$ standard deviation and independent-samples $t$-test was used in the statistical analysis. $P < 0.05$ was considered as significant difference. * indicated $P < 0.05$; ** indicated $P < 0.01$.

## RESULTS

### Differentially expressed genes in RA

Nine mRNA expression profiles of RA synovial tissues, including 168 patients with RA and 41 healthy individuals, were incorporated into our study (Table 1). In our analysis, a total of 378 DEGs including 202 up- and 176 down-regulated genes were identified in patients with RA compared to healthy individuals. LCK and MAT2A were the most significantly up- and down-regulated genes in RA, respectively (Table 2). The expression pattern of the

**Table 1** The expression profiling of patients with RA and normal controls.

| GEO ID | Sample (N:P) | Platform | Year | Author | Country |
|---|---|---|---|---|---|
| GSE77298 | 7:16 | GPL570[HG-U133_Plus_2] Affymetrix Human Genome U133 Plus 2.0 Array | 2016 | Mathijs G.A. Broeren | Netherlands |
| GSE48780 | 0: 83 | GPL570[HG-U133_Plus_2] Affymetrix Human Genome U133 Plus 2.0 Array | 2014 | Sarah K. Kummerfeld | USA |
| GSE55235 | 10:10 | GPL96[HG-U133A] Affymetrix Human Genome U133A Array | 2014 | Thomas Häupl | Germany |
| GSE55457 | 10:13 | GPL96[HG-U133A] Affymetrix Human Genome U133A Array | 2014 | Raimund W. Kinne | Germany |
| GSE55584 | 0:10 | GPL96[HG-U133A] Affymetrix Human Genome U133A Array | 2014 | Woetzel D | Germany |
| GSE36700 | 0:7 | GPL570[HG-U133_Plus_2] Affymetrix Human Genome U133 Plus 2.0 Array | 2012 | Bernard Robert Lauwerys | Belgium |
| GSE24742 | 0:12 | GPL570[HG-U133_Plus_2] Affymetrix Human Genome U133 Plus 2.0 Array | 2010 | Bernard Robert Lauwerys | Belgium |
| GSE12021 | 9:12 | GPL96[HG-U133A] Affymetrix Human Genome U133A Array | 2008 | René Huber | Germany |
| GSE1919 | 5:5 | GPL91[HG_U95A] Affymetrix Human Genome U95A Array | 2004 | Ungethuem U | Germany |

**Notes.**
RA, rheumatoid arthritis; N, normal control; P, patients.

top 200 DEGs in RA patients and healthy individuals was shown in Fig. 1, which indicated that the mRNA expression profiling of synovial tissues of RA patients was different from healthy individuals.

## GO classification of DEGs in RA

DEGs in patients with RA were performed to GO annotation for investigating the biological roles. The threshold of the GO terms was set as FDR < 0.001. As Table 3 shown, RNA splicing (GO: 0007585), signal transduction (GO: 0007165) and gene expression (GO: 0010467) were the highly significant enrichment of the GO biological process. Furthermore, protein binding (GO: 0005515), nucleotide binding (GO: 0000166) and ATP binding (GO: 0005524) were the highest enrichment of the GO molecular function.

## Pathway enrichment analysis

KEGG analysis was used to understand the signaling pathway enrichment of DEGs in RA. The threshold was FDR < 0.05. As Table 4 shown, the significant enrichment of pathways focused on axon guidance (hsa04360), RNA transport (hsa03013), protein processing in endoplasmic reticulum (hsa04141) and MAPK signaling pathway (hsa04010).

## Protein-protein interaction network

The PPI network between the top 10 up- and down-regulated DEGs in RA were constructed by Cytoscape based on the BioGRID database (Fig. 2). The network consisted of 419 nodes and 480 edges. In the network, the nodes with high degree were defined as hub proteins. The hub proteins in the PPI network including RBFOX2 (degree = 90), LCK (degree = 78), SERBP1 (degree = 45), TCOF1 (degree = 36) and TARS (degree = 32), as Fig. 2

**Table 2** The top 10 up- and down-regulated DEGs in RA.

| Gene ID | Gene Symbol | $p$-Value | FDR | Regulation |
|---|---|---|---|---|
| **Top 10 up-regulated DEGs** | | | | |
| 3932 | LCK | 0 | 0 | Up |
| 6964 | TRD | 1.65645E–13 | 4.7104E–10 | Up |
| 26135 | SERBP1 | 1.95311E–10 | 3.33239E–07 | Up |
| 8711 | TNK1 | 2.47189E–09 | 3.01253E–06 | Up |
| 7456 | WIPF1 | 3.19709E–08 | 2.72744E–05 | Up |
| 5209 | PFKFB3 | 3.67501E–08 | 2.85014E–05 | Up |
| 23543 | RBFOX2 | 6.50956E–08 | 3.7022E–05 | Up |
| 56681 | SAR1A | 7.13709E–08 | 3.80541E–05 | Up |
| 961 | CD47 | 8.72732E–08 | 4.13626E–05 | Up |
| 9669 | EIF5B | 9.36124E–08 | 4.20319E–05 | Up |
| **Top 10 down-regulated DEGs** | | | | |
| 4144 | MAT2A | 1.27898E–13 | 4.7104E–10 | Down |
| 1388 | ATF6B | 9.32052E–11 | 1.98783E–07 | Down |
| 26148 | C10orf12 | 6.51139E–10 | 9.25812E–07 | Down |
| 6949 | TCOF1 | 5.24047E–09 | 5.58831E–06 | Down |
| 2516 | NR5A1 | 8.37334E–09 | 7.937E–06 | Down |
| 6897 | TARS | 4.75819E–08 | 3.12247E–05 | Down |
| 5463 | POU6F1 | 4.41294E–08 | 3.12247E–05 | Down |
| 10412 | NSA2 | 5.26449E–08 | 3.20795E–05 | Down |
| 763 | CA5A | 8.50556E–08 | 4.13626E–05 | Down |
| 23387 | SIK3 | 1.4221E–07 | 5.77712E–05 | Down |

**Notes.**
RA, rheumatoid arthritis; DEGs, differentially expressed genes; FDR, false discovery rate.

shown. A total of 402 predicted proteins interaction with top 10 up- and down-regulated were overlapped with 202 up- and 168 down-regulated DEGs, respectively. Lastly, 15 DEGs interacted with top 10 up- and down-regulated DEGs were obtained and their interactions were shown in Table S2. RBFOX2 interacted with 6 dysregulated DEGs including DDX5, ATN1, ESR2, RPA2, RRPA3 and ATXN2; LCK interacted with WASL; SERBP1 interacted with PIAS1; TCOF1 interacted with RPA2, RPA3, TOP1 and NOP56.

## Construction of transcription factors regulatory network

In our study, TFs regulatory network under the regulation of TFs and target genes were constructed. Based on the TRANSFAC database, total of 42 proteins were identified as TFs of top 10 up- and down-regulated DEGs in RA. The identified TFs and target genes were applied to construct the TFs-target genes network by Cytoscape software. As shown in Fig. 3, the network consisted of 61 nodes and 142 edges. The DEGs, such as RBFOX2 (degree = 16), POU6F1 (degree = 11), WIPF1 (degree = 10), PFKFB3 (degree = 9) and EIF5B (degree = 9) had the high connectivity with TFs.

## The expression levels of DEGs were analyzed in the GSE17755 datasets

GSE17755 covered the expression profiling of peripheral blood cells from 112 RA patients and 45 health individuals. The descriptive statistics (spread and distribution) of DEGs

Hao et al. (2017), *PeerJ*, DOI 10.7717/peerj.3078

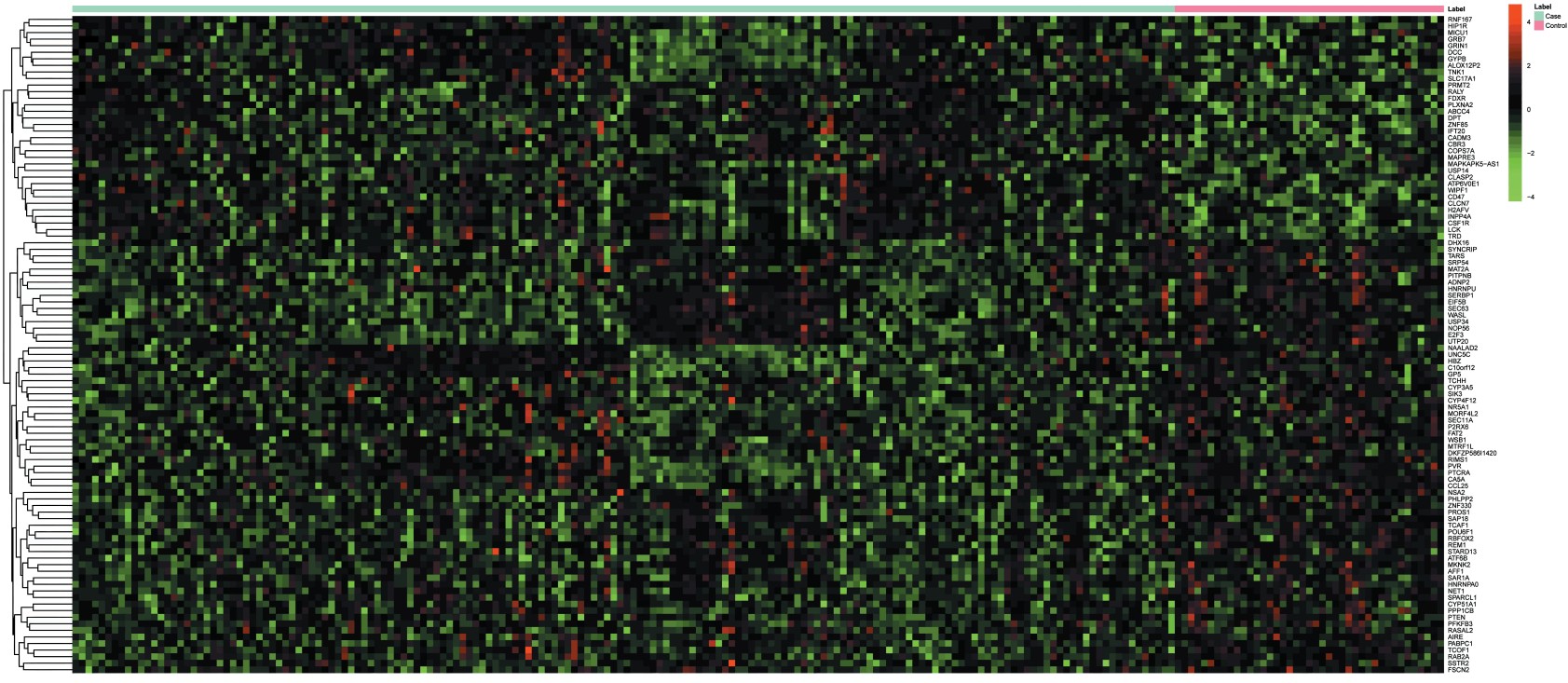

**Figure 1** **Hierarchical clustering analysis based on the expression profile of the top 200 discriminatory DEGs between synovial tissues of RA and normal controls.**
The color scale illustrated the relative expression level of mRNA across all samples; red color represented an expression level above mean, green color represented expression lower than the mean.

**Table 3  GO annotation of dysregulated DEGs in RA.**

| GO Items | GO terms | Genes | FDR |
|---|---|---|---|
| **Biological process** | | | |
| GO:0008380 | RNA splicing | 19 | 9.89E–08 |
| GO:0007165 | Signal transduction | 35 | 2.39E–05 |
| GO:0010467 | Gene expression | 19 | 3.43E–05 |
| GO:0016477 | Cell migration | 10 | 5.30E–05 |
| GO:0006397 | mRNA processing | 13 | 5.31E–05 |
| GO:0000398 | Nuclear mRNA splicing, via spliceosome | 12 | 5.64E–05 |
| GO:0045944 | Positive regulation of transcription from RNA polymerase II promoter | 21 | 0.00018566 |
| GO:0046777 | Protein autophosphorylation | 10 | 0.00062775 |
| GO:0006355 | Regulation of transcription, DNA-dependent | 37 | 0.00104687 |
| GO:0070934 | CRD-mediated mRNA stabilization | 3 | 0.00127416 |
| **Molecular Function** | | | |
| GO:0005515 | Protein binding | 162 | 4.77E–48 |
| GO:0000166 | Nucleotide binding | 66 | 3.56E–13 |
| GO:0005524 | ATP binding | 45 | 3.29E–08 |
| GO:0003677 | DNA binding | 47 | 7.84E–07 |
| GO:0003723 | RNA binding | 22 | 8.07E–05 |
| GO:0003676 | Nucleic acid binding | 25 | 0.000137 |
| GO:0003724 | RNA helicase activity | 4 | 0.000148 |
| GO:0003713 | Transcription coactivator activity | 12 | 0.000202 |
| GO:0003700 | Sequence-specific DNA binding transcription factor activity | 26 | 0.000207 |
| GO:0004497 | Monooxygenase activity | 7 | 0.000214 |

Notes.
DEGs, differentially expressed genes; FDR, false discovery rate; RA, rheumatoid arthritis.

expression in RA patients compared to healthy individuals were depicted by using box-plot analysis, which was visually illustrated by median and inter-quartile range. As Figs. 4F, 4G and 4K shown, the expression levels of MAT2A and NSA2 were significantly down-regulated and CD47 had the up-regulated tendency in peripheral blood cells of patients with RA, which were consistent with our integrated analysis in RA synovial tissues. However, the expression of LCK, SERBP1, WIPF1, PFKFB3, RBFOX2, ATF6B, POU6F1 and CA5A in GSE17755 was incompatible with our integrated analysis in RA synovial tissues (Figs. 4A–4E, 4H–4J).

## qRT-PCR validation of the expression levels of candidate genes in RA

qRT-PCR was subjected to validate the expression levels of dysregulated genes in peripheral blood samples of 10 RA patients and 10 healthy individuals. Three candidate DEGs including MAT2A, NSA2 and CD47 were used to apply for qRT-PCR verification. As Figs. 5A and 5B shown, MAT2A ($P < 0.01$) and NSA2 ($P < 0.05$) were obviously down-regulated in peripheral blood samples of RA patients compared with healthy individuals. However, the expression level of CD47 ($P > 0.05$) in peripheral blood samples was not

**Table 4   KEGG enrichment of dysregulated DEGs in RA.**

| KEGG ID | KEGG terms | FDR | Genes |
|---|---|---|---|
| hsa04360 | Axon guidance | 0.010565 | DCC, SEMA6A, SEMA4D, PLXNA2, RGS3, NFAT5, CDK5, UNC5C |
| hsa03013 | RNA transport | 0.015948 | EIF5B, SAP18, GEMIN4, PABPC1, THOC2, EIF1B, EIF3E |
| hsa00982 | Drug metabolism - cytochrome P450 | 0.017355 | ALDH1A3, CYP1A2, CYP3A5, MGST3, UGT2B15 |
| hsa00980 | Metabolism of xenobiotics by cytochrome P450 | 0.017377 | ALDH1A3, CYP1A2, CYP3A5, MGST3, UGT2B15 |
| hsa05200 | Pathways in cancer | 0.018693 | PPARD, PIAS2, DCC, E2F3, PTEN, PDGFA, CSF1R, MAPK8, PIAS1, BRCA2, CEBPA |
| hsa03040 | Spliceosome | 0.020842 | DHX15, HNRNPA1, DHX16, THOC2, PRPF3, HNRNPU, SRSF5 |
| hsa04910 | Insulin signaling pathway | 0.021514 | RPS6KB1, MAPK8, MKNK2, TSC1, PHKA1, PRKAG1, PPP1CB |
| hsa05016 | Huntington's disease | 0.021575 | POLR2J, NDUFA2, UQCRC1, TGM2, BBC3, NDUFA7, GRIN1, ATP5O |
| hsa05010 | Alzheimer's disease | 0.023516 | NDUFA2, UQCRC1, CAPN2, NDUFA7, GRIN1, ATP5O, CDK5 |
| hsa04141 | Protein processing in endoplasmic reticulum | 0.024488 | SEC63, P4HB, SAR1A, CAPN2, MAPK8, AMFR, HERPUD1 |
| hsa03060 | Protein export | 0.02558 | SRP54, SEC63, SEC11A |
| hsa04010 | MAPK signaling pathway | 0.026502 | MAPK8IP2, PDGFA, RRAS, MAPK8, MAP3K11, MKNK2, DUSP3, STK3, CACNB2, MAP3K8 |
| hsa03440 | Homologous recombination | 0.031046 | RPA2, RPA3, BRCA2 |
| hsa05160 | Hepatitis C | 0.033309 | PIAS2, MAPK8, PIAS1, IFNA21, IRF9, EIF3E |

**Notes.**

DEGs, differentially expressed genes; FDR, false discovery rate; RA, rheumatoid arthritis.

significant difference between RA patients and healthy individuals, but its expression displayed the up-regulated tendency in peripheral blood samples of RA patients.

## DISCUSSION

Rheumatoid arthritis, a chronic inflammatory disease of joint, is triggered by the complex interaction between genetic susceptibility and dysregulated genes. The clinical manifestations of RA patients present pannus and the abnormal proliferative synovial tissue. Early diagnosis would improve the prognosis and the life quality of RA patients. However, the pathogenesis mechanism of RA is ambiguous and there are no feasible biomarkers for early diagnosis in clinical practice. In our study, bioinformatics analysis-based identification of key genes and pathways in RA were conducted, aiming to provide valuable ground work for further investigation of pathogenesis mechanism of RA.

RBFOX2 belongs to binding protein, is thought to be a key regulator of alternative exon splicing in the nervous system and other cell types (*Venables et al., 2013a*). It was one

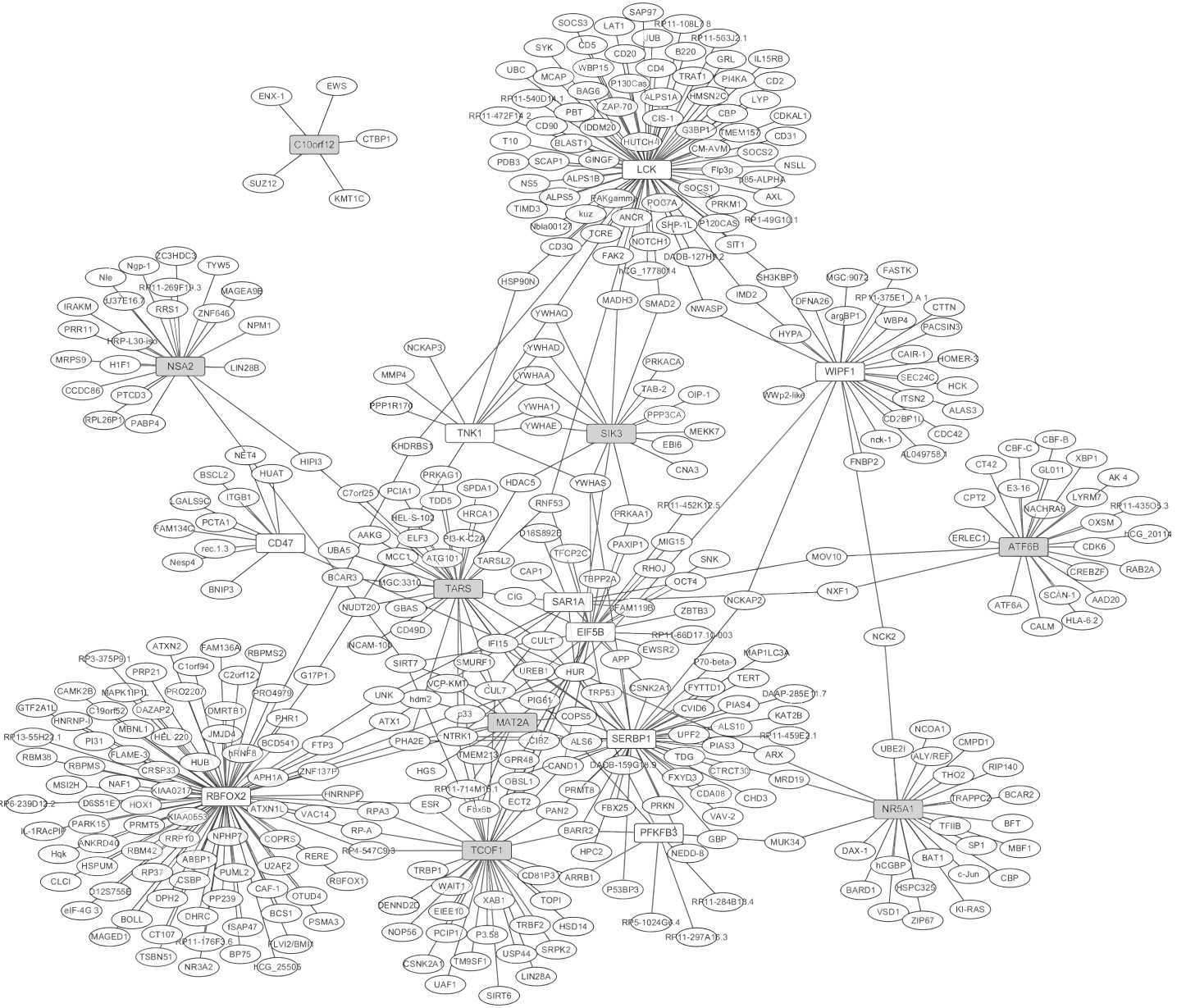

**Figure 2** **The protein-protein interaction network of the top 10 up- and down-regulated DEGs in synovial tissues of RA.** White and grey rectangle nodes represented up- and down-regulated DEGs, respectively. White ellipse nodes denoted products of genes predicted to interact with the DEGs. The solid line indicated the interaction correlation between proteins.

of top 10 up-regulated genes in RA synovial tissues (Table 2). In the TFs regulatory network, RBFOX2 had the highest connectivity with TFs, was regulated by 16 TFs. RBFOX2 is involved in various diseases and biological processes including congenital heart disease, breast cancer, epithelial-mesenchymal transition and pluripotent stem cell differentiation (*Homsy et al., 2015*; *Braeutigam et al., 2014*; *Cen et al., 2013*; *Venables et al., 2013b*). The mutations of RBFOX2 are associated with congenital heart disease and neuro developmental disabilities (*Homsy et al., 2015*). During the epithelial-mesenchymal
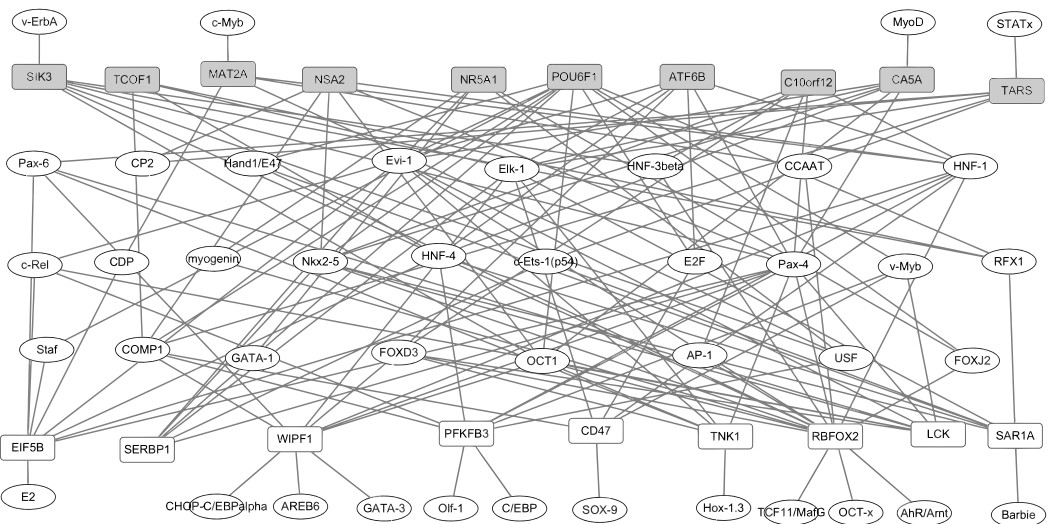

**Figure 3  The transcription factors-target genes regulatory network of top 10 up and down-regulated DEGs in synovial tissues of RA.** Transcription factors can regulate gene expression through binding to the promoter regions of target genes. The white and grey rectangle nodes represented up- and down-regulated DEGs, respectively. The ellipse nodes represented transcription factors. Solid lines represented regulatory correlation between transcription factors and DEGs. For instance, NSA2 was down-regulated in synovial tissues of RA and its expression level can be regulated by six transcription factors including CP2, COMP1,Nkx2-5, OCT1, Pax4 and HNF-1 based on TRANSFAC database prediction.

transition, RBFOX2 is an essential regulator to mediate cellular invasion (*Braeutigam et al., 2014*). To our knowledge, our study first reported the aberrant expression of RBFOX2 in RA, however, the biological function of RBFOX2 in RA needs to be further elucidated.

*CD47* encodes CD47 molecule, a membrane protein, is involved in the increase of intracellular calcium concentration and may play an essential role in membrane transport and signal transduction (*Kaur et al., 2015*). CD47 was significantly down-regulated in RA (Table 2). In the TFs regulatory network, CD47 was targeted by 6 TFs including v-Myb, SOX-9, HNF-1, COMP-1, E2F and Elk-1. In RA, the interaction between CD47 and thrombospondin-1 could trigger T cell infiltration and expansion in the rheumatoid synovium, and perpetuates the inflammatory process in the rheumatoid joint (*Vallejo et al., 2003*). CD47 might play essential roles in the progression of RA. In our study, the expression level of CD47 was increased in synovial tissues and peripheral blood cells of patients with RA (Fig. 4F).

*MAT2A* encodes methionine adenosyltransferase 2A, is an enzyme that catalyzes the production of S-adenosylmethionine (AdoMet) from methionine and ATP, and is a key methyl donor in cellular processes (*Panayiotidis et al., 2006*). In our study, MAT2A was the most significantly down-regulated DEG in synovial tissues of RA (Table 2). The previous studies display that MAT2A is associated with uncontrolled cell proliferation in cancer. The protein expression level of MAT2A is decreased in renal cell carcinoma compared to normal tissues (*Wang et al., 2014*), while MAT2A is over-expressed in various gastrointestinal cancers, such as gastric cancer, colon cancer and liver cancer (*Zhang et al., 2013*; *Tomasi et al., 2013*). miR-21-3p targets down-regulation of MAT2A and inhibits

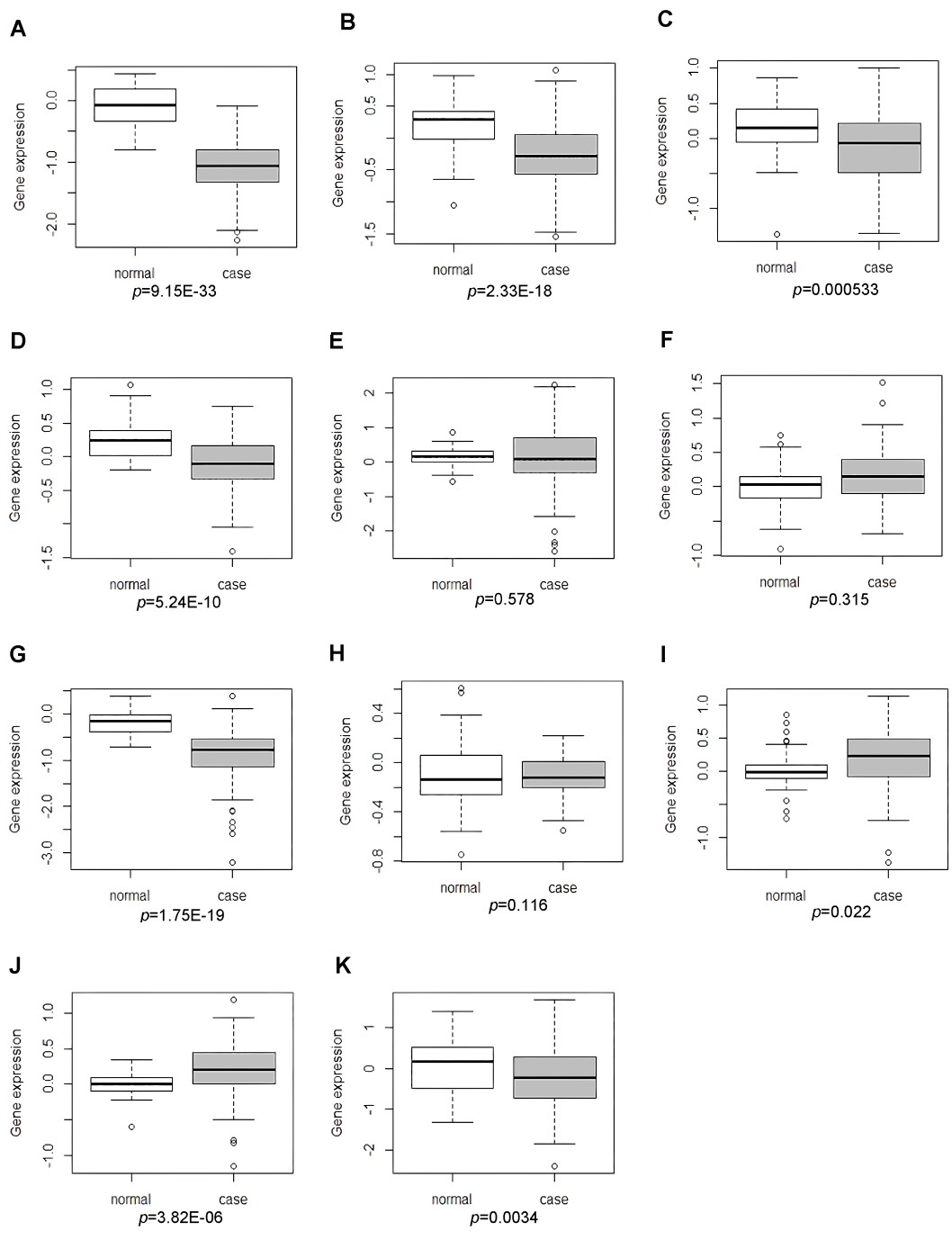

**Figure 4** **The expression levels of DEGs was analyzed in the GSE17755 dataset.** Box-plot diagram was used to describe the median and inter-quartile range of DEGs expression including LCK (A), SERBP1 (B), WIPF1 (C), PFKFB3 (D), RBFOX2 (E), CD47 (F), MAT2A (G), ATF6B (H), POU6F1 (I), CA5A(J) and NSA2(K).

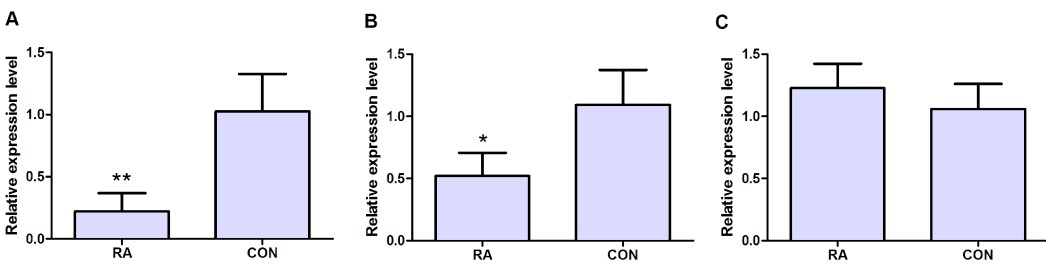

**Figure 5** **qRT-PCR validation of candidate DEGs in peripheral blood samples of RA patients and healthy individuals.** (A) The expression level of MAT2A; (B) the expression level of NSA2; (C) the expression level of CD47. RA represented rheumatoid arthritis and CON represented healthy individuals. * represented $P < 0.05$ and ** represented $P < 0.01$.

cell growth in hepatoma (*Lo, Tsai & Chen, 2013*). Based on aforementioned information, down-regulated MAT2A in synovial tissues of RA might implicate in cell proliferation, invasiveness of fibroblast-like synoviocytes, which results in erosion of bone and cartilage in RA. The expression of MAT2A in synovial tissues and FLS needs to be validated through quantitative real-time polymerase chain reaction in a larger sample size of patients with RA and healthy individuals, in addition, the biological roles of MAT2A in FLS needs to be further investigated through *in vivo* and *in vitro* experiment. MAT2A had the lower expression in both of peripheral blood cells and synovial tissues of RA patients compared to healthy individuals (Fig. 4G).

NSA2 encodes NSA2 (Nop seven-associated 2), ribosome biogenesis homolog, locates in the nucleolus of the cell, and is involved in cell cycle regulation and proliferation (*Zhang et al., 2010*). It had the lower expression level in synovial tissues and peripheral blood cells of RA patients compared to healthy individuals (Table 2, Fig. 4K). Over-expression of the NSA2 protein promotes cell growth and regulates the G1/S transition in the cell cycle in different cell lines including HeLa (human cervical carcinoma), HEK293T (human embryonic kidney) and A549 (human epithelial lung adenocarcinoma) (*Zhang et al., 2010*; *Li et al., 2013*). The association between dysregulated NSA2 and pathogenesis of RA has not been reported. The functions of NSA2 in initiation and progression of RA need to be elucidated in the future study.

Dysregulated DEGs were identified in synovial tissues of patients with RA, and the mRNA expression level of representative DEGs was preliminarily detected in peripheral blood cells of patients with RA based on GSE17755 dataset. As Fig. 4 shown, LCK, SERBP1, WIPF1, PFKFB3 and RBFOX2 were down-regulated in peripheral blood cells of RA, but those genes were up-regulated in synovial tissues of RA; POU6F1 and CA5A were down-regulated in peripheral blood cells of RA, but they were up-regulated in synovial tissues of RA. The contradictory expression status of DEGs in synovial tissues and peripheral blood cells might attribute to different characteristics of expression profiling between synovial tissues and peripheral blood cells in patients with RA. In our bioinformatics analyses, the expression levels of MAT2 and NSA2 were obviously down-regulated both in synovial tissues and peripheral blood samples of RA patients; and CD47 expression was up-regulated both in synovial tissues and with the up-regulated tendency in peripheral blood samples of RA
patients. The expression status of MAT2, NSA2 and CD47 in peripheral blood samples of RA patients were validate through qRT-PCR. The qRT-PCR results indicated MAT2A and NSA2 were significantly down-regulated in peripheral blood samples of RA patients and CD47 had the up-regulated tendency in peripheral blood samples of RA patients compared with healthy individuals (Fig. 5). In general, qRT-PCR results of MAT2A, NSA2 and CD47 were compatible with our bioinformatics analyses. The associations between a majority of identitied TFs in the TF-gene regulatory network and RA remain unknown. The roles of a number of identified TFs in other autoimmue disorders are explored. c-Ets-1 belongs to the ETS family of transcript factors and regulates the expression of various immune-related genes. In our work, it was predicted to bind with the promoter of SAR1A, SERBP and TARS. Over-expression of c-Ets-1 in synovial membrane regulates inflammatory angiogenesis in RA through being activated by interleukin-1 and tumor necrosis factor alpha (*Wernert et al., 2002*; *Redlich et al., 2001*). A eQTL analysis indicates that the allele of rs6590330 in ETS1 is associated with decreased ETS1 expression and increases the risk of systemic lupus erythematosus by enhancing the binding of pSTAT1 (*Lu et al., 2015*). The regulatory significance of identified TFs in development and progression of RA needs to be profoundly investigated in our future work.

The dysregulated genes in RA were significantly enriched in KEGG pathways including axon guidance, RNA transport and MAPK signaling pathway. Axon guidance represents a key stage in the formation of neuronal network. Axons are guided by a variety of guidance factors, such as netrins, ephrins, slits, and semaphorins. Catalano reports that semaphoring-3A (SEMA3A), a number of semaphorin family, attenuates inflammation and progression of collagen-induced arthritis (*Catalano, 2010*). SEMA3A is decreased in the immune system of experimental autoimmune encephalomyelitis murine model of multiple sclerosis, which is one of autoimmunity diseases (*Gutierrez-Franco et al., 2016*). In our study, down-regulated SEMA6A and up-regulated SEMA4D were significantly enriched in axon guidance pathway. THOC2 (interacted with NR5A1 in PPI network) and EIF58 (one of the top 10 up-regulated DEGs in RA) were significantly enriched in RNA transport pathways. The mitogen-activated protein kinase (MAPK) cascade is involved in various cellular functions, including cell proliferation, differentiation and migration. A series of articles depict that MAPK signaling pathway associates with RA (*Pearson & Jones, 2016*; *Weisbart et al., 2013*; *Schett, Zwerina & Firestein, 2008*). Increased expression of phosphorylation ezrin may contribute to migration and invasion of fibroblast-like synoviocytes in RA, which are mediated by rho kinase and the p38 MAPK pathway phosphorylation (*Xiao et al., 2014*). BRAF splice variants activate MAPK through CRAF, increase expression of MT1-MMP, and enhance fibroblast invasion of collagen in RA. Based on the abovementioned information, axon guidance, RNA transport and MAPK signaling pathway might contribute to the progression of RA.

In our study, RBFOX2 and those dysregulated DEGs interacted with it including DDX5, ATN1, ESR2, RPA2, RPA3 and ATXN2 were significantly enriched in six significant enrichment of biological process and molecular function, such as RNA splicing, signaling transduction, gene expression, protein binding , nucleotide binding and ATP binding (Table S1). NPM1 (interacted with NAS2) was significantly enriched in signal transduction

and protein binding (Table S1). Our finding indicated that the protein-protein interactions between dysregulated DEGs might play essential roles in progression and development of RA through those biological processes and molecular functions.

There are limitations in our study. Firstly, further biological function studies should be performed to investigate the roles of these DEGs and pathways in the progression of RA. Secondly, whether those dysregulated genes in RA have the diagnostic value in RA diagnosis needs to be validated through large cohorts of RA patients and healthy individuals.

## CONCLUSIONS

In conclusion, we identified 378 DEGs in synovial tissues of RA patients compared to healthy individuals. The top 10 up- and down-regulated genes were selected to establish the PPI network and TF-target genes network. The expression levels of candidate DEGs was preliminarily detected in peripheral blood cells of RA according to GSE17755 dataset. Our study indicated that dysregulated genes in RA might contribute to the progression of RA by regulating biological process, molecular function and signaling pathways based on complex gene regulatory network interaction, such as PPI network and TF-genes regulatory network. Our findings may pave the road for exploring the underlying mechanisms of RA and identifying the potential biomarkers for RA diagnosis.

### Funding
The authors received no funding for this work.

### Competing Interests
Haiwei Du is an employee of Beijing Medintell Biomed Co., Ltd. Ruihu Hao and Haiwei Du equally contribute to the work and are co-first authors.

### Author Contributions
- Ruihu Hao and Haiwei Du performed the experiments, analyzed the data, contributed reagents/materials/analysis tools, wrote the paper, prepared figures and/or tables, reviewed drafts of the paper.
- Lin Guo conceived and designed the experiments, wrote the paper, prepared figures and/or tables, reviewed drafts of the paper.
- Fengde Tian and Ning An performed the experiments, reviewed drafts of the paper.
- Tiejun Yang and Changcheng Wang analyzed the data, reviewed drafts of the paper.
- Bo Wang and Zihao Zhou contributed reagents/materials/analysis tools, reviewed drafts of the paper.

### Data Availability
NCBI Gene Expression Omnibus database: GSE77298, GSE48780, GSE55235; GSE55457, GSE55584, GSE36700, GSE24742, GSE12021, GSE1919, GSE17755.

## Supplemental Information

Supplemental information for this article can be found online at http://dx.doi.org/10.7717/peerj.3078#supplemental-information.

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
