# Peer review of "Identification of dysregulated genes in rheumatoid arthritis based on bioinformatics analysis"

_PeerJ, doi:10.7717/peerj.3078_

## Round 0.1 · original submission · Major Revisions

The manuscript has been carefully evaluated by three external reviewers and their comments are attached. We found that the research question is relevant and of interest, however there are many methodological issues that need to be properly addressed. The manuscript as it currently stands will thus require substantial major revisions to address all the concerns raised by the reviewers. In particular, we suggest a revision by a native English speaker of the final text. Additional analyses and calculations are required in order to: i) build a proper statistical framework for differential expression (DE) analyses, ii) to link the protein-protein network and DE findings, iii) to better exploit the association between transcription factors and DE genes, iv) to discuss in details the candidates genes in a more biological/pathway context, v) to tune down some of the claims on 'discover of biomarkers for early detection, vi) a better usage and grouping of the GO terms.

I would be glad to consider a major revised version of this manuscript in which all the points raised by the reviewers will be properly addressed one by one.

Reviewer 1 ·

Basic reporting

The paper by Hao et al. seeks to pinpoint novel biomarkers for early diagnosis of rheumatoid arthritis (RA) via bioinformatic analysis of open source data. The authors utilize microarray expression profiles from RA patients and healthy controls to identify biomarker candidates through differential expression (DE) analysis, protein-protein interaction networks and other computational methods. After identification of genes found to be DE in synovial tissue between RA samples and controls, the authors go on to evaluate the potential of these genes as RA diagnostic-markers through comparison of gene expression in RA peripheral bold cells.
While the research question is very relevant and interesting the manuscript itself lacks a proper work-through tying introduction, discussion and future perspectives together. Additionally, some methodology should be explained in better detail and most importantly, a major “clean up” of the language is needed.

1. The language requires a good deal of work. Many places prepositions are lacking and generally the language deteriorates as the article progresses. Some sentences are very difficult to understand, one example is line 229-232.

2. References are needed at the end of lines 51, 196, 205, 207 and 220.

3. The article fits PeerJ standards except abstract headings which should have periods instead of colons: “Headings in structured abstracts should be bold and followed by a period. Each heading should begin a new paragraph”.

Experimental design

1. Figure 2, containing the protein-protein interaction network would be a lot more relevant/interesting with some commenting on the interacting genes from the bioGRID database. That is, are any of these also differentially expressed (DE) and are they DE in the same direction (up, down) as the hub gene they interact with.

2. The methods section on association between transcription factors from TRANSFAC and differentially expressed genes should be explained in more detail, that is, what does “screening” entail?

3. Elaboration on the 10 genes found to be most significantly up/down regulated. Are these genes chosen purely based on FDR (info on this is lacking)? Additionally, perhaps consider using a log fold change (LogFC) in addition to FDR (or if nothing else report it for better overview).

4. The results of the gene ontology enrichment could be better represented, perhaps through grouping of GO-terms. Vague categories should be removed, like that of “cellular component” which confers no information in this case.

Validity of the findings

No Comments

Additional comments

Why use box plots instead of another DE analysis when analyzing the peripheral blood and comparing it to the synovial tissues? Is it because you speculate that the signal in the blood may be to weak for the DE-analysis to pick up or are there other reasons? It would be nice to have same type of analyses performed on both sets of data, making the results more straight forward to compare.

Reviewer 2 ·

Basic reporting

- The submitted paper seems adhere to all PeerJ Policies.
- Although in general the written English is acceptable, an additional English edition is recommended.
- The format of the submitted article conform the templates requested by PeerJ

Experimental design

The authors conducted a gene expression study in synovial tissues from patients with rheumatoid arthritis (RA). The raw data was obtained from databases deposited in open access repositories online. The main finding of this study is the identification of genes differentially expressed between cases and controls, plus potential biological relationships deduced from in silico analysis. Some of the top-ranked genes were analyzed in an additional dataset obtained from a different type of sample (peripheral blood cells), from RA patients and controls.

There are methodological, biological and clinical inconsistencies:

1) Authors stated in the inclusion criteria that “…RA patients received none treatment before collection of tissue 81 samples”. However, patients from dataset GSE24742 (published by Gutierrez-Roelens et al. DOI: 10.1002/art.30292) were all declared as anti-TNF-resistant RA patients, implying that all of them were treated with an immunomodulatory drug previous to the extraction of tissue sample. Similarly, the 10 samples contained in the dataset GSE55584 (published by Woezel et al. DOI 10.1186/ar4526) were obtained from RA patients with medication (different types of drugs) at the moment of sampling (see article).
Therefore, the inclusion criteria was not conducted in the way the authors say.

2) No major conclusions were obtained from the Gene Ontology, PPI, TFs and Pathway analyses. The authors identified ontologies related to RNA splicing, signal transduction and gene expression, among other general terms. All this terms are expected when you are working with gene expression data. The GO terms are meant to give biological meaning to expression data and also lead or highlight the next experimental steps. However, the results obtained from GO and pathway say nothing relevant about the biology behind the expression signature obtained.

3) The main conclusion of this paper is that CD47, MAT2A and NSA2 are potential biomarkers for early diagnosis of RA. This observation was obtained from the analysis of top-ranked mRNAs in peripheral blood cells of patients with RA and controls. However, the analysis on peripheral blood cells showed that levels of NSA2 and CD47 are fairly similar between cases and controls and given that the authors did not even provide a statistical analysis it is not possible to conclude that there is a true difference in mRNAs levels.

4) Importantly, biomarkers should be related to the disease in a relevant manner. Synovial tissues are composed mainly by macrophage-like synovial cells and fibroblasts. Under inflammatory processes, such as RA, an increase in the immune infiltration may be seen. On contrary, peripheral blood cells are composed by myeloid and lymphocytic lineages. What it is the rationale of searching synovial tissue-derived mRNAs in blood cells? How these genes are related to the biology of synovial tissues? What types of cells are expressing these mRNAs?

For instance, LCK and TRD were the two top-ranked genes in synovial tissues. However, the expression of LCK is importantly reduced in peripheral blood cells. LCK and TRD are two genes strongly associated with T-cell lymphocytes, however, the authors did not address the biological relevance of finding both genes up-regulated in synovial tissues and LCK strongly down-regulated in peripheral blood cells.

5) The authors stated that these markers may be used for “early detection” of RA. All the datasets contained patients with a positive diagnostic of RA at the moment of sampling (an important proportion with medication). Early detection implies the identification of the disease in subjects without clinical symptoms, however, samples from subjects obtained prior to diagnostic were not included in this analysis. This proposition is not supported by the results

Specific comments:

Materials and methods
Line 83: the authors stated “…Respective 4 datasets were based on sequencing platform of GPL570 [HG-U133_Plus_2] 84 Affymetrix Human Genome U133 Plus 2.0 Array and GPL96 [HG-U133A] Affymetrix Human 85 Genome U133A Array, 1 dataset was based on platform of GPL91 [HG_U95A] Affymetrix 86 Human Genome U95A Array”

Comment: All these codes are for Affymetrix genechips based on microarray technology, no sequencing. What do you mean with “sequencing platform” in this sentence? Please clarify

Figure 1
Question: What it is the purpose of figure 1? Datasets and Cases/Controls are splitted al over the heatmap not showing any particular pattern. Were patients clustered?. For instance, the dataset GSE48780 contains only patients (comprising the 50% of patients in the present study) and seems to split into two different expression patterns. Do the authors have an explanation for this?

Legend: “…red color represents an expression level above mean, blue color represents expression lower than the mean”
Comment: The heatmap was built using red and green colors, what do you mean with blue color? There is no blue color in the heatmap, please amend.

Figure 4
Authors did not provide a statistical analysis. This is particularly relevant for CD47 and NSA2 where the differences in expression are not obvious and both mRNAs represent the main conclusion of this paper.

Validity of the findings

- The main conclusions obtained by the authors are not supported by the results.
- There are serious flaws in the experimental design, analysis and conclusions of this study.
- The study has an excessive emphasis on bioinformatic analysis and does not consider seriously the biology and clinical features of the disease. By no means it is clear how the bioinformatic analysis of GO, pathways, PPI and TFs contribute to establish a rationale for the diagnosis of RA.

Additional comments

No comments

Reviewer 3 ·

Basic reporting

There are several places within this manuscript where the written English is poor and therefore the standard unacceptable for publication in its current format. For example, the entire paragraph beginning from line 52 to 62. This is highlighted again in the following paragraph - lines 63-68. This hampers the comprehension of the text. Furthermore, I am concerned by the lack of the reporting of any genome wide association findings (GWAS) data in relation to RA. Instead the authors highlight case control studies - the first one cited being from a Slovakia population.
Within the introduction, where the authors mention environmental risk factors it is again surprising that no mention is made either here, nor later in the discussion, of epistasis.

Experimental design

The experimental design as presented is acceptable but the findings can at best be seen as preliminary. The authors provide no explanation as to why only GSE 17755 datasets have been utilised. Has ethnicity, gender, RA disease duration been controlled for within this dataset?
The most significant weakness in the experimental design is the lack overall of any integration with any genetic information. In order for the TRANSFAC analysis to be meaningful this should have been aligned to replicated SNP loci co-segregating with the gene expression data.

Validity of the findings

Whilst the findings may indeed be valid my concern is that these findings, as they stand, are highly preliminary and can therefore lead to a misdirection of the potential biomarkers. This can be robustly addressed by factoring in data from replicated SNP loci.

Additional comments

Causal analysis through the utilisation of network approaches is an advantageous way forwards for understanding complex genetic diseases. The use of gene expression data sets coupled with the TRANSFAC approach is useful but limited in the way currently presented. The manuscript also requires careful attention with regards to written English for clarity throughout.

---

## Round 0.2 · Minor Revisions

The manuscript has been substantially improved. There are still some issues to solve especially highlighted by the reviewer #3 before endorsing this manuscript for publication. We encourage the authors to carefully address them one by one.

Reviewer 2 ·

Basic reporting

The clarity and objectivity of the article has improved since its previous edition, particularly in the methodological description.

Experimental design

Methodological errors were corrected and flaws were improved

Validity of the findings

Although methodological errors were corrected, the biological relevance of the findings is uncertain. However, it may help to support RA research.

Additional comments

Thank you for making the effort to correct and improve your study.

Reviewer 3 ·

Basic reporting

The improvements made by Hao et al to the English have improved the overall clarity of the manuscript. However, I do not feel that the authors build a comprehensive background. They have now added GWAS findings but this appears in a somewhat random manner. There is no understanding of replicated loci post GWAS, nor of the importance of eQTLs https://www.ncbi.nlm.nih.gov/pubmed/25954001; https://www.ncbi.nlm.nih.gov/pubmed/26404118. Similarly, within the Discussion the authors now include reference to SNP findings in relation to certain of the TFs. I am not convinced by the lists of associations that have been grouped together in this way. This is a concern as it may prove to over state their findings to the reader.
Attempting to incorporate the SNP details as suggested, therefore, has exposed a weakness in the authors' comprehension. These parts of the current manuscript need to be re-written.
Greater detail should be added into the figure legend for Figure 3. as some of the regulatory correlations seem difficult to comprehend following the current description.

Experimental design

The way in which the Data collection (para 2) of the Material & Methods is written is confusing. 9 mRNA were utilised. There is a 4 missing within this listing.

Validity of the findings

The findings are on the whole weak. Independent qRT PCR of 2 or 3 loci would substantially improve the validity of the data.

---

## Round 0.3 · accepted · Accept

All the previous concerns expressed by the reviewers have been now addressed and resulted in a solid manuscript. I am glad to accept this article for publication in PeerJ.

Reviewer 3 ·

Basic reporting

As previously reported

Experimental design

An appropriate number of RA cases and controls have been examined for 3 index genes by qRT-PCR

Validity of the findings

The new experimental data support the bioinformatic analyses

Additional comments

The additional experimental work which has now been included together with the improved clarity and content of the text has resulted in a significant elevating of the manuscript.